# Low-Complexity Acoustic Scene Classification Using Time Frequency Separable Convolution

**Duc H. Phan** *,† and **Douglas L. Jones** †

Electrical and Computer Engineering, University of Illinois at Urbana-Champaign, Champaign, IL 61801, USA
* Correspondence: ducphan2@illinois.edu
† These authors contributed equally to this work.

**Abstract:** Replacing 2D-convolution operations by depth-wise separable time and frequency convolutions greatly reduces the number of parameters while maintaining nearly equivalent performances in the context of acoustic scene classification. In our experiments, the models' sizes can be reduced by 6 to 14 times with similar performances. For a 3-class audio classification, replacing 2D-convolution in a CNN model gives roughly a 2% increase in accuracy. In a 10-class audio classification with multiple recording devices, replacing 2D-convolution in Resnet only reduces around 1.5% of the accuracy.

**Keywords:** low complexity audio network; acoustic scene classification; depth-wise separable convolutions; detection and classification of acoustic scenes and events

## 1. Introduction

Acoustic scene classification (ASC) has become an important research topic in recent years [1–3]. The goal of ASC is to categorize given recordings into a set of given classes.

Deep neural networks are currently the best performing techniques in a wide range of applications in computer vision, bioinformatics, medical disease study, robotics, and audio processing applications [4]. Many network structures and architectures have first been developed in the context of computer vision or image classification problems and later adapted to other application domains. Acoustic scene classification, which classifies sound recordings into a set of predefined classes, has also adopted variants of the two dimensional Convolutional Neural Network (CNN) from computer vision in recent studies [5–8]. Variants of VGG [9], ResNet [10] and DenseNet [11] are state of the art architectures from image classification which have been successfully applied to acoustic scene classification problems [5–8,12]. Similar to the networks used in computer vision [13], millions of parameters are often required for applying deep neural networks in acoustic scene classification problems [7]. Such large networks require a lot of computational power for training, and present challenges for deployment on mobile phones or low-power-consumption devices. As a consequence, low-complexity neural network solutions are a topic of great interest in acoustic classification applications. In this study, we propose a method to decompose a traditional 2D convolution operation [14] into a series of small convolution operators in order to design a low complexity neural network for acoustic scene application.

In order to design a smaller network for a given task, one can start by training a large network for performance, and then train a network of similar structure with fewer parameters to match the output of the original network in the training set [15]. Alternatively, pruning a network zeros out a large fraction of the network parameters to reduce complexity [16–18]. In pruning schemes, a low-complexity network is a model which has a small number of non-zero parameters within the original complex network structure. A pruned network can be achieved by iteratively zeroing out a small fraction of parameters having lower magnitude and then retraining the model until a desired compression ratio is reached [16,18].

In contrast, designing network structures that inherently have only a small number of parameters directly reduces the computational cost in both training and inference. Mobilenets [13,19] are examples of networks that can reduce the number of parameters required while maintaining reasonable performance. Key features of these networks include separable convolutions, depth-wise separable convolutions, and linear bottlenecks [19,20]. Our study aims at finding a low complexity architecture for audio. The key idea is to recognize the distinction between a 2D spectrogram of an audio clip, in which the time and frequency dimensions represent fundamentally different characteristics, whereas both dimensions in a 2D image are spatial translation. For example, in image classification an image and its transpose would normally be classified as the same class, while this would rarely be the case for a spectrogram and its transposed version.

Our approach shares some similarities with EEGNet-based networks [21,22] which are low-complexity networks used in BMI (Brain-Computer Interface) applications. EEGNet classifies Electroencephalography (EEG) signals by using multiple 1D convolutions along temporal and spatial dimensions in the different layers of the network. The success of EEGNets in providing the low-computational-complexity networks with high accuracy in BMI applications supports our idea to exploit 1D convolutions to design low-complexity networks in acoustic scene applications. There are two main differences from our approach and EEGNets. Firstly, our study focuses on acoustic scene classification from audio recordings, which have their own distinct characteristics. Secondly, our approach tries to build a decomposition of 2D convolution into a series of 1D convolutions, and then apply the decomposition into high-performance deep neural networks in different acoustic scene applications.

Our main contributions of this paper are: first, time-frequency separable convolution is introduced to decompose 2D convolution for acoustic scene classification. Secondly, we show how to apply time-frequency separable convolution into a given network structure in order to reduce the number of parameters significantly while maintaining similar performances. In our experiments, we can reduce total parameters by 14 times for a simple CNN and more than 6 times for a complex Resnet.

In this study, we demonstrate our contributions through low-complexity architectures on the dataset from DCASE 2020 task 1 subtask B [8] and DCASE 2021 task 1 subtask A [23]. For the DCASE 2020 task 1 dataset, we extend the baseline network from the DCASE 2020 [8] to work with binaural audio as our baseline for comparison. Meanwhile, in the DCASE 2021 dataset, as our baseline we selected the much more complex Residual network solution [24] that had a high performance on the dataset. To make the trade-off clear we limited the architecture changes, and our solutions are mainly achieved by replacing 2D-convolution operations in the baseline networks with our proposed decomposition.

The rest of this paper is organized as follows: First, a description of time-frequency separable convolution is provided before introducing the datasets for the experiments. Next, for each experiment, each dataset is explained. Each proposed network is described after outlining the corresponding baseline network. Lastly, a discussion of the experimental results is followed by the conclusion.

## 2. Time-Frequency Separable Convolution

When applying a deep neural network to acoustic scene classification, a spectrogram of an audio clip is treated as a 2D image and fed into a deep convolutional neural network. From the sample spectrogram in Figure 1, we can see that the frequency and time axes are not interchangeable for each other; therefore simply applying 2D convolutional operations for audio spectrum input fails to exploit the unique characteristics of the audio domain. As a result, we propose the time-frequency separable convolution structure.

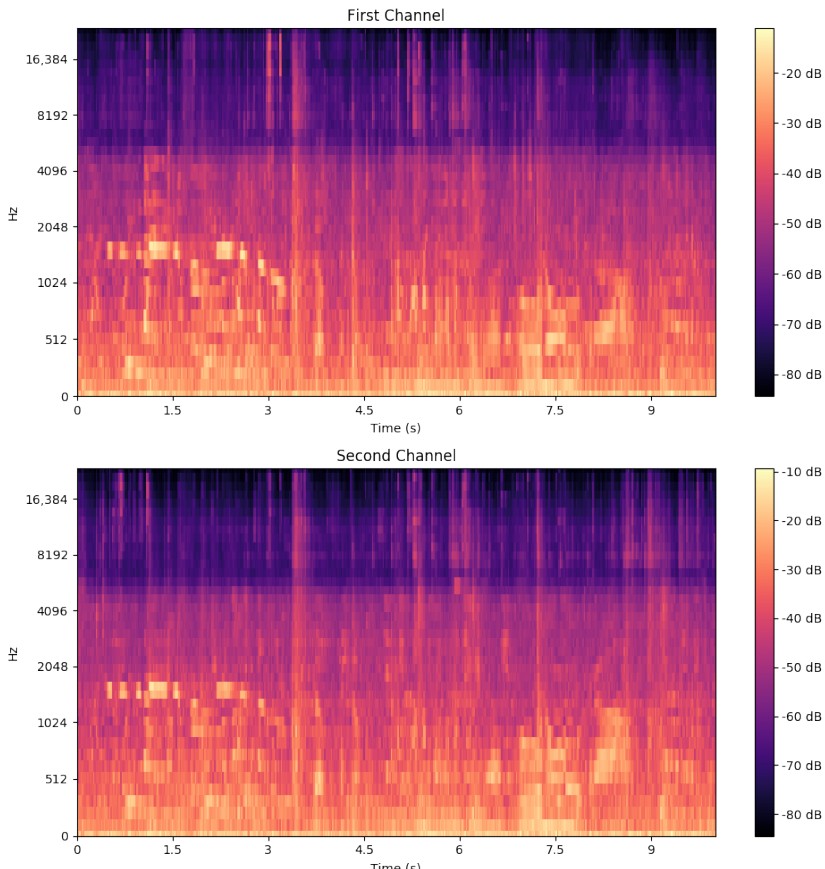

**Figure 1.** Sample spectrogram from the DCASE 2020 Task 1 Subtask A development dataset. This clip is recorded at an airport. The spectrogram of each channel from the binaural recording is depicted.

Before diving into details of the proposed structure, we can review 2D convolution in an audio context as shown in Figure 2. Given a multiple-channel two dimensional input P, the output at the given frequency bin $f$, time $t$ is defined as

$$O(f,t) = \sum_{i=1}^{C_f} \sum_{j=1}^{C_t} \sum_{k=1}^{C_{in}} w_{ijk} p_{f+i-1,t+j-1,k} \tag{1}$$

where $p_{ijk}$ are input at frequency $i$, time step $j$, and channel $k$. $w_{ijk}$ is the corresponding weight of the convolution layer. $C_{in}$ is the number of input channels to the convolution layer, while $C_t$ and $C_f$ specify the size of the convolution operator. Note that $p_{ijk}$ is set to zero if $i$ or $k$ is larger than the number of frequencies and time steps of the input, respectively.

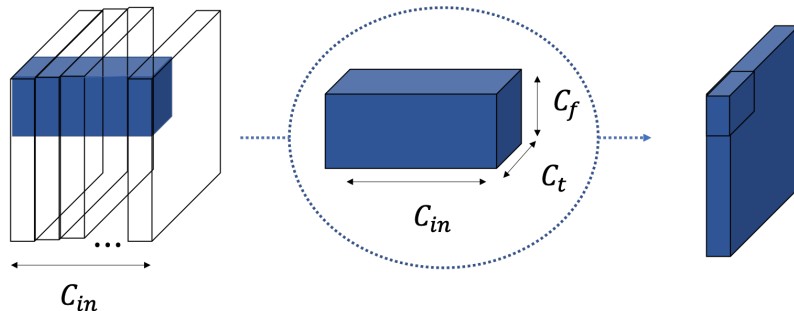

**Figure 2.** 2D convolution operation illustration at a given location of an output layer.

The 2D convolution can be decomposed into the separable time-frequency structure by a two-step process. First, each channel of the input is convolved along the frequency axis before applying convolution along the time axis as shown in Figure 3. Equations (2) and (3) describe these steps in detail

$$F(f, j, k) = \sum_{i=1}^{C_f} w_{i,k}^f p_{f+i-1,j,k} \tag{2}$$

$$T(i, t, k) = \sum_{j}^{C_t} w_{j,k}^t F(i, t+j-1, k) \tag{3}$$

where $w_{j,k}^f$ and $w_{j,k}^t$ are the parameters of the convolution along the frequency and time axes of input channel $k$, respectively. Secondly, outputs from the frequency convolutions and time convolutions are concatenated to form intermediate input $I(i, j, k)$ as defined in Equation (4). Input $I$ is fed to a $1 \times 1$ convolution as depicted in Figure 4. Equation (5) expresses the final output of the time-frequency separable convolution.

$$I(i, j, k) = \begin{cases} F(i, j, \frac{k}{2}) & \text{if } k \mod 2 = 0. \\ T(i, j, \frac{k+1}{2}) & \text{otherwise.} \end{cases} \tag{4}$$

$$O(f, t) = \sum_{k=1}^{2C_i n} w_k I(f, t, k) \tag{5}$$

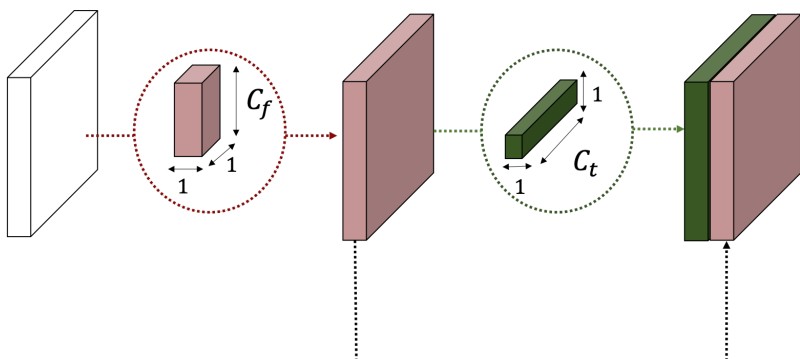

**Figure 3.** The first step of 2D convolution decomposition. In this step the convolution along frequency is applied, followed by convolution along the time axis.

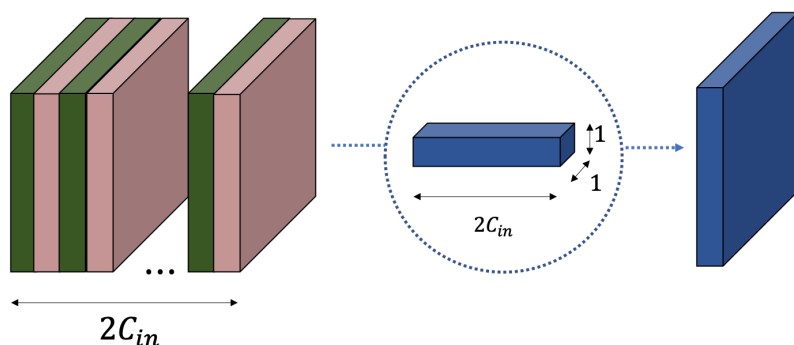

**Figure 4.** The second step of 2D convolution decomposition. In this step $1 \times 1$ convolution is applied to generate the final output layer.

The proposed convolution structure decomposes the representation into two separable time and frequency components, thereby reducing the number of parameters compared

to the traditional 2D convolution. For example, if a hidden convolution layer of a neural network has $C_{in}$ input channels and $C_{out}$ output channels, where the size of convolution is $C_f$ and $C_t$, then the total parameters of 2D convolution $C(C_{in}, C_{out}, C_f, C_t)$ is given by Equation (6)

$$C(C_{in}, C_{out}, C_f, C_t) = C_{in} \times C_{out} \times C_f \times C_t \tag{6}$$

while building a similar hidden layer using our time-frequency separable convolution requires the number of parameters $C^{tf}(C_{in}, C_{out}, C_f, C_t)$ provided by Equation (7)

$$C^{tf}(C_{in}, C_{out}, C_f, C_t) = 2C_{in} \times C_{out} + C_{in} \times (C_f + C_t)$$
$$= C_{in} \times (2C_{out} + C_f + C_t) \tag{7}$$

As a consequence, the compression ratio is provided by Equation (8)

$$\frac{C(C_{in}, C_{out}, C_f, C_t)}{C^{tf}(C_{in}, C_{out}, C_f, C_t)} = \frac{C_{in} \times C_{out} \times C_f \times C_t}{C_{in} \times (2C_{out} + C_f + C_t)}$$
$$= \frac{C_{out} \times C_f \times C_t}{2C_{out} + C_f + C_t} \tag{8}$$

For example, if a hidden layer has $C_{in} = 64$, $C_{out} = 64$, $C_f = 5$, and $C_t = 5$, then by applying Equations (6) and (7), the number of parameters for traditional convolution and the proposed convolution are 102,400 and 8832. Hence, the compression ratio in terms of parameters required is roughly 11.5 times in the given example.

The time-frequency separable convolution structure can easily be extended to increase its representations. For example, a non-linear activation function can be applied to the output of certain stages in the time-frequency convolution structure. In our implementation, batch normalization [25] and rectified linear units (Relu) [25] are applied before $1 \times 1$ 2D convolution. Furthermore, when the proposed structure is applied to the input of a model, a 2D convolution of size $C_{f \times 1}$, which is a convolution along the frequency axis, should be chosen for the model to learn extra low-level features in the frequency dimension. Because of the flexibility of the proposed structure, the total number of parameters in some practical implementations can be generalized by adding extra positive term $\alpha$ as shown in Equation (9).

$$C^{tf}(C_{in}, C_{out}, C_f, C_t) = C_{in} \times (2C_{out} + C_f + C_t + \alpha) \tag{9}$$

## 3. Experiment with CNN-Based Network

In the first experiment, we apply our time-frequency architecture in a Convolutional Neural Network (CNN). Our design goal is a small model for acoustic scene classification. We select the DCASE 2020 Task 1 Subtask B dataset for this experiment because the dataset was created for low complexity model developments for acoustic scene classification in the DCASE 2020 challenge. In addition, the baseline CNN for this task is used as our base network for model size reduction.

### 3.1. Dataset

The DCASE 2020 Task 1 subtask B dataset contains recordings of 10 different acoustic scenes from 12 European cities [8]. The acoustic scenes are grouped into three classes: indoor, outdoor, and transportation. All recordings are binaural, 48 kHz 24-bit format, and from a single recording device. The development data set has 40 h of recording from 10 different cities. 70 percent of the development dataset is used for training, while 30 percent is withheld as a test set. Recordings from the same location appear only in the training set or test set but not both. In audio applications, raw audio is typically transformed into log-mel energy spectrograms, with the machine learning algorithm operating on these log-mel energy features. Sample spectrograms of a recording from the dataset are depicted in Figure 1. Note that the DCASE 2020 Task 1 subtask B dataset has an equal number of

instances of each class. We maintain the balance between the classes in the training set and test set during our experiments.

### 3.2. CNN Baseline Architecture

The baseline network is a convolutional neural network consisting of the input layer followed by three hidden layers and an output layer. Hidden layers include two convolution layers followed by a fully-connected layer. The first convolution layer has 32 filters of size $7 \times 7$, while the second hidden layer contains 64 filters. Conv2D(32,7,7) and Conv2D(64,7,7) denote the first two hidden layers. The fully connected layer FC(100) has 100 neurons, followed by a fully connected output layer FC(3) that has the same number of classes as the dataset. Rectified linear units (Relu) [25] are used as activation functions in hidden layers, while Softmax [25] is applied at the output of the network. In the hidden layers, batch normalization units are applied before Relu units. Furthermore, max pooling [25] layers MaxP(5,5) and MaxP(4,100) are connected to the end of each convolution layer, respectively. Max pooling MaxP(5,5) has pool size of $5 \times 5$, while layer MaxP(4,100) uses the pool size of $4 \times 100$. Dropout layers Dropout(0.3) are applied in all hidden layers. The dropout rate in the baseline network is 0.3. The baseline network is similar to the baseline network from DCASE 2020 on the dataset except that the input layer in our baseline network accepts two Mel spectrogram channels of a recording. Table 1 summarizes the base network structure. The total number of parameters of the baseline network is roughly 117 K. The proposed network will modify and apply our time-frequency separable convolution in order to reduce the number of parameters significantly. Our proposed network will be discussed in detail in the following section.

**Table 1.** Summary of the baseline network in terms of connection configurations and numbers of parameters.

| Layer | Number of Parameters |
|---|---|
| Input | 0 |
| Conv2D(32,7,7) | 3136 |
| Batchnorm | 128 |
| Relu | 0 |
| MaxP(5,5) | 0 |
| Dropout(0.3) | 0 |
| Conv2D(64,7,7) | 100,352 |
| Batchnorm | 256 |
| Relu | 0 |
| MaxP(4,100) | 0 |
| Dropout(0.3) | 0 |
| FC(100) | 12,900 |
| Batchnorm | 400 |
| Relu | 0 |
| Dropout(0.3) | 0 |
| FC(3) | 303 |
| Softmax | 0 |
| Output | 0 |
| Total parameters | 117,475 |

### 3.3. Proposed Architecture

The proposed network is designed by first replacing all of the 2D convolutions in the baseline by our time-frequency separable structure. Instead of a fully-connected hidden layer, we apply 2D global max pooling to reduce each input channel into one single maximum value. For acoustic scenes classification, we think that persistent features are more reliable than transient features; therefore, average pooling [25] is employed instead of max pooling layers. Furthermore, in a final version, a smaller filter size is used, and the positions of pooling layers are changed compared to the baseline model. The details

of the architecture can be described as follows. Average pooling across time of size $1 \times 5$ is first applied to the input of the model. Next, the separable time-frequency structure $\mathrm{Conv}_{tf}(32,4,5)$ with filter lengths of four and five along frequency and time, respectively, replaces the first convolution layer of the baseline network. The output of the time-frequency structure has 32 channels that are average-pooled by size $2 \times 3$. The time-frequency structure $\mathrm{Conv}_{tf}(64,5,5)$ generates 64 output channels, followed by a global max-pooling layer. The frequency and time filters of the second time-frequency structure have lengths of 5. Lastly, the output of the model includes three fully connected units. Relus are used as the activation functions in hidden layers, while softmax is applied at the output layer. In addition, batch normalization units are added at outputs of time-frequency structures. Table 2 summarizes the proposed network. Our network only uses 8k parameters, which is roughly 14 times smaller than the baseline network. If each parameter is 32 bit floating point, our model only requires 32 KB for storage while the baseline model needs nearly 470 KB as shown in Table 3.

**Table 2.** Summary of the proposed network in term of connection configurations and number of parameters.

| Layer | Number of Parameters |
|---|---|
| Input | 0 |
| AverageP(1,5) | 0 |
| $\mathrm{Conv}_{tf}(32,4,5)$ | 2980 |
| Batchnorm | 128 |
| Relu | 0 |
| AverageP(2,3) | 0 |
| $\mathrm{Conv}_{tf}(64,5,5)$ | 4672 |
| Batchnorm | 256 |
| Relu | 0 |
| GlobalMaxPooling | 0 |
| FC(3) | 195 |
| Softmax | 0 |
| Output | 0 |
| Total parameters | 8003 |

**Table 3.** Model sizes of the baseline network and the proposed architecture in KB when each parameter is a 32 bit floating point.

| System | Number of Parameters | Total Size |
|---|---|---|
| Baseline | 117,475 | 469.9 KB |
| Proposed Structure | 8003 | 32 KB |

### 3.4. Experiment Setup

The baseline system and the proposed architecture are trained and evaluated on the aforementioned dataset. Once again, this experiment is designed to evaluate the performance impact of the time and frequency separability in CNN for audio applications, so the changes are restricted to these convolutional components. In the experiment, each audio channel of a recording is converted to a log-mel-band energy spectrogram with 40 mel bands. The number of samples in an analysis frame is 2048 (40 ms) with 50% hop interval. The log-mel-band energy spectrogram features of the recordings are normalized frequency-wise across time step by mean and standard deviation from the training set before inputting into the studied models. In other words, the dataset $D$ has the form given by Equation (10).

$$D = \{X | X \in \mathbf{R}^{F \times T \times C}\} \tag{10}$$

where $F$ is 40 mel bands and $T$ is 498 analysis frames (10 s audio recording) and $C$ is 2 channels of the binaural recordings.

During training, 30 percent of the training set is withheld for validation. Each model is trained for 200 epochs with a batch size of 64. The ADAM optimizer [26] was used with a learning rate of 0.0001. The parameter was learned by minimizing the cross-entropy loss function given by Equation (11)

$$L_\theta = -\frac{1}{N} \sum_i^N Y_i \cdot log(f_\theta(X_i)) \tag{11}$$

where $N$ is number of training examples, vectors $X_i$ and $Y_i$ represent training example $i$ and its corresponding ground truth, respectively. $f_\theta(\cdot)$ is the machine learning model. Note that dot product is used for the multi-class classification problem in Equation (11) and $Y_i$ is one-hot encoding in our case. The parameter values of a model are selected such that the classification error over the validation set is smallest among all epochs.

Because our model is very compact compared to the baseline in terms of the number of parameters, we therefore think that data augmentation is more suitable as a regularization technique than dropout. Therefore the mix-up data augmentation technique [27] was explored for the proposed model during one of the training processes. Mix-up generates weighted combinations of random pairs of audio recordings from the training data. Given two recordings and their ground truths $(X_i, Y_i)$, $(X_j, Y_j)$, a synthetic training example $(\hat{X}, \hat{Y})$ is given by Equation (12)

$$\begin{aligned} \hat{X} &= \lambda X_i + (1 - \lambda) X_j \\ \hat{Y} &= \lambda Y_i + (1 - \lambda) Y_j \end{aligned} \tag{12}$$

where $\lambda$ is sampled from the $Beta(\alpha = 0.2, \beta = 0.2)$ distribution [28] independently at the beginning of each epoch when the mix-up technique is used.

*3.5. Performance*

The performance on the dataset is measured on the validation subset. Our primary metric is the classification accuracy. Accuracy is calculated as a macro-average: average of the class-wise accuracy for the acoustic scene classes. More precisely, if the number of predefined classes is $C$, and $N^{(c)}$ samples belong to class $c$, then the accuracy metric is given by Equation (13)

$$\text{Accuracy} = \frac{1}{C} \sum_c^C \frac{1}{N^{(c)}} \sum_{i:y_i=c} I\{y_i = \hat{y}_i\} \tag{13}$$

where $I$ is the identity function (true when predicted label $\hat{y}$ matches label $y_i$). In addition, we also measure multi-class cross-entropy (log loss) as a metric which is independent of the operating point [8]. The multi-class cross-entropy is shown in Equation (11). Each model was trained and tested 10 times; the means and standard deviations of the performance from these 10 independent trials are shown in the result Table 4.

**Table 4.** Performance of the models on DCASE2020 Task 1b dataset.

| System | Accuracy (%) | Log Loss |
|---|---|---|
| Baseline model | $88.96 \pm 0.56$ | $0.352 \pm 0.064$ |
| Proposed model | $90.15 \pm 0.77$ | $0.293 \pm 0.024$ |
| Proposed model with mixup | $91.14 \pm 0.40$ | $0.287 \pm 0.006$ |

Clearly, the proposed network using time-frequency convolution outperforms the baseline system even though it is much smaller in size. The accuracy of the baseline system on average is 88.96% while the corresponding number for the proposed network is 90.15%. When mix-up data augmentation is employed during the training of the proposed architecture, the average performance increases by roughly 1% to 91.14%. Mix-up data augmentation also helps to reduce the performance variance of independent trials. In

terms of log loss metric, the proposed model with mix-up shows the smallest loss, while the baseline has the largest loss on average. Overall, the proposed model with mix-up data augmentation gives the best result in both accuracy and log loss metric even though the model size is around 14 times smaller than the baseline system. Figure 5 presents the class-wise accuracy for different experimental settings. Without mix-up data augmentation during training, the proposed architecture only shows improvement for detecting the outdoor class while the accuracy of indoor and transportation classes is similar to the baseline network. When mix-up is employed, the proposed model increases in accuracy in all classes compared to the baseline network. Furthermore, in each class the proposed network with mix-up has the smallest standard deviation.

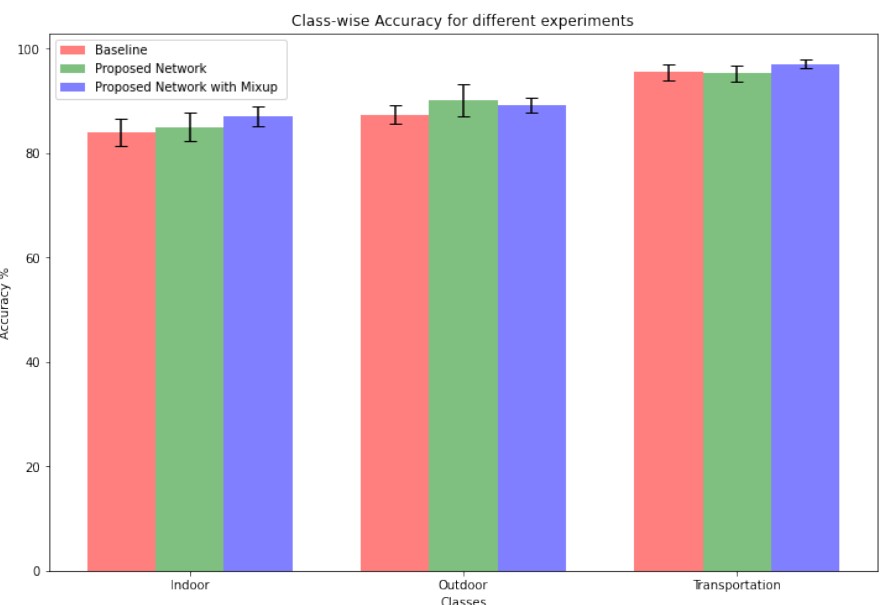

**Figure 5.** Class-wise accuracy for different experiments with the DCASE 2020 Task 1A dataset. The proposed network is constructed by replacing 2D convolution in the baseline network with time-frequency separable convolution.

## 4. Experiment with Resnet Based Network

### 4.1. Dataset

The DCASE 2021 Task 1 subtask A dataset contains recordings of 10 different acoustic scenes from 12 European cities with 4 recording devices [8]. From the original recording devices, 11 simulated devices are created by applying different impulse responses and dynamic compression ranges from recordings of Device A. The development dataset includes three real Devices A, B, and C, and six simulated Devices S1–S6. In addition, the development dataset only includes recordings from 10 cities. The acoustic scenes are grouped into 10 classes: airport, shopping mall, metro station, pedestrian street, public square, street traffic, tram, bus, metro, and park. 64 h of 24-bit format recordings of single-channel audio at a sampling rate of 44.1 kHz are provided in the development dataset. This dataset has similar number of samples of each class. During our experiments, we try to maintain an equal number of instances for each class in the training and test set.

The baseline prepossessing steps convert each recording to log mel-band energy spectrograms with 128 mel bands. The number of samples in an analysis frame was 2048 with 50% hop interval. Each spectrogam was normalized into a range from 0 to 1 by its maximum and minimum values. Log-mel deltas and delta-deltas without padding were included as additional inputs into the models. The un-normalized version of an input audio recording from the dataset is given in Figure 6.

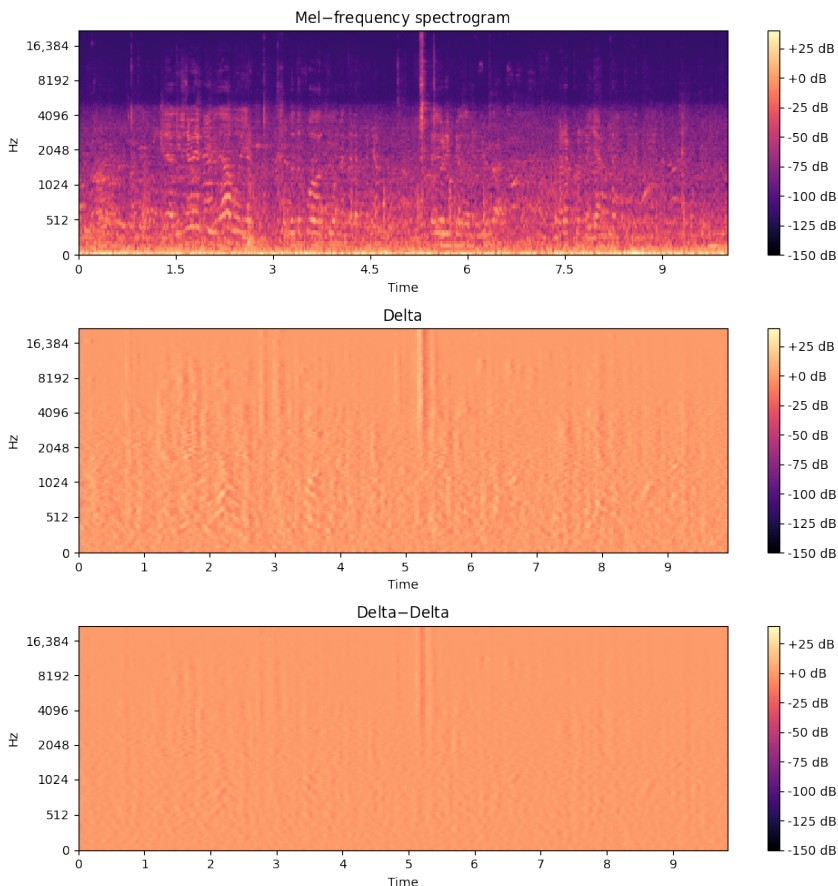

**Figure 6.** Sample spectrogram from the DCASE 2021 Task 1 Subtask A development dataset. This clip was recorded at an airport in Barcelona.The delta and delta-delta features are presented along with the spectrogram feature.

### 4.2. ResNet Baseline Model

Our baseline model was a smaller version of the ResNet model from [24]. The key elements of the Resnet structure are the Residual Blocks. The ResNet baseline relies on two types of residual blocks: ResBlock1 and ResBlock2 as shown in Figure 7. In both residual blocks, the input processes through two $3 \times 3$ 2D convolutions before an addition operation on one path. ResBlock1 has a skip connection directly from the input of the block to the addition operation at the output, while ResBlock2 modifies the input $x$ by an average pooling operation over a $3 \times 3$ window. Note that ResBlock1(16) denotes 16 2D convolution filters in each 2D convolutional layer of the ResBlock1 block while ResBlock2(32) represents 32 2D convolution filters in each 2D convolutional layer of the ResBlock2 block.

In the ResNet baseline model, the spectrogram inputs are split evenly in mel bins into two parts $X_{low}$ and $X_{high}$. Each part is first fed into an independent branch including 2D convolution, ResBlock1s, and ResBlock2s. Outputs of the two branches are concatenated before applying two $1 \times 1$ 2D convolutions in series as shown in Figure 8. Note that batch-normalization is applied before Relu inside ResBlock1 and Resblock2. The 2D convolutions in this baseline implicitly have batch-normalization and Relu at their outputs.

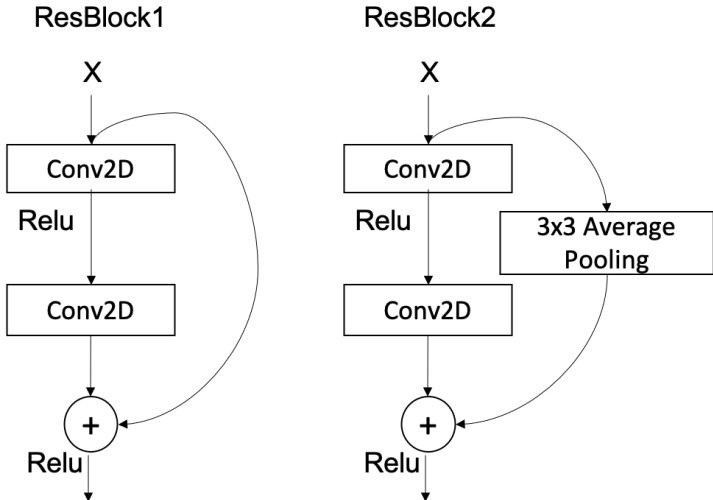

**Figure 7.** Two types of residual blocks are used in the baseline ResNet. ResBlock1 has the skip connection directly from the input of the block to the addition operation at the output, while ResBlock2 modifies the input $x$ by an average pooling operation over a $3 \times 3$ window.

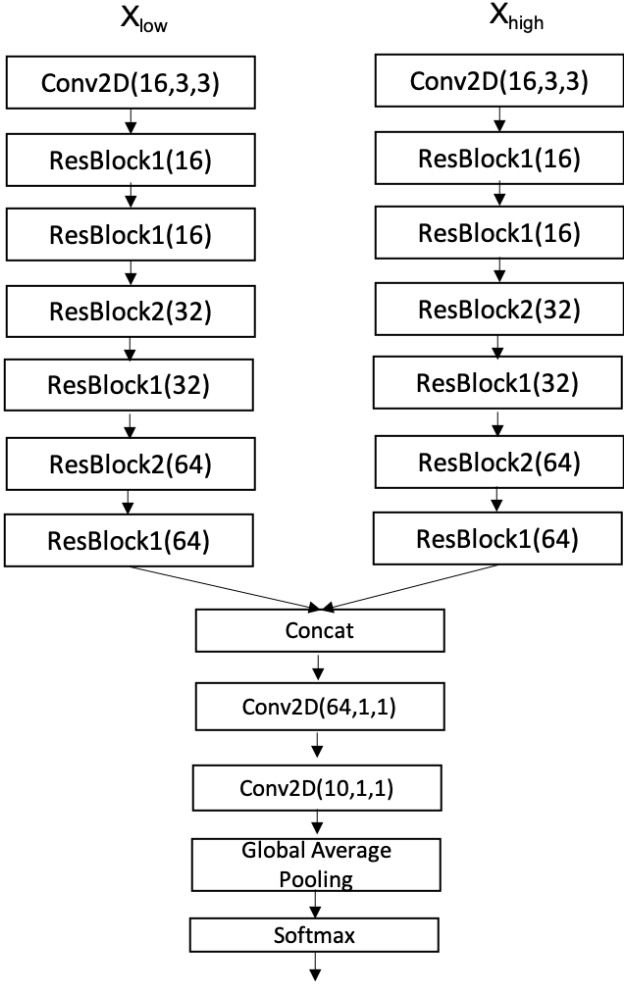

**Figure 8.** The ResNet baseline. The input is split into independent vectors by the mel frequency bins. The top 50% of the high frequencies form $X_{high}$, and the remaining mel bins belong to $X_{low}$. Each vector goes through a 2D convolution before connecting to the chain of residual blocks. Outputs from the chain of residual blocks are concatenated before running though two 2D Convolution operations to produce the output for the classification.

### 4.3. Compressed-ResNet Model

In the compressed ResNet, we apply the time-frequency separable convolution to replace 2D convolution inside residual blocks ResBlock1 and ResBlock2. Namely, the proposed ResBlock1 and ResBlock2 are shown in Figure 9. Note that $5 \times 5$ convolution windows are used in the first and the second $\text{Conv}_{tf}$s respectively in the residual blocks. The architecture of the final network still follows closely the baseline ResNet in Figure 8 except that all ResBlock1s and ResBlock2s are replaced by their corresponding compressed version. The number of parameters reduced by a factor of 6.4 when the time-frequency separable convolutions are applied as shown in Table 5.

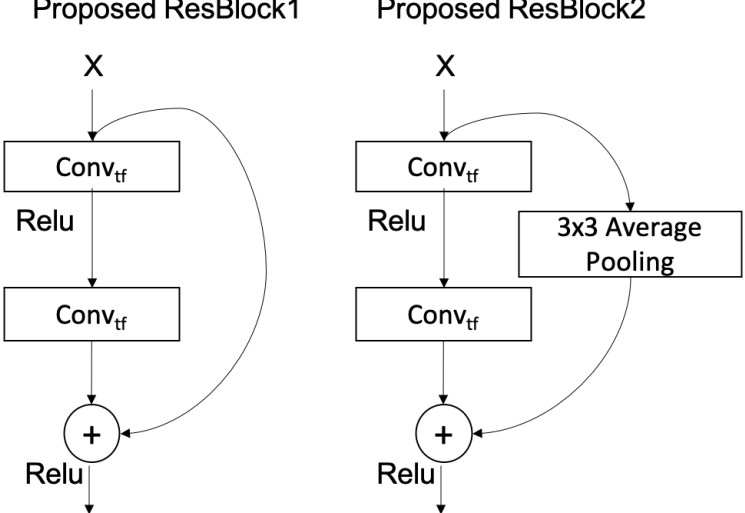

**Figure 9.** Compressed ResBlock1 and Compressed ResBlock2 are composed of replacing 2D convolution with time-frequency separable convolution.

**Table 5.** Summary model parameters for ResNet networks.

| Model | Total Number of Parameters |
| :---: | :---: |
| Baseline ResNet | 363,084 |
| Compressed ResNet | 57,484 |

### 4.4. Experiment

The experiments with the ResNet models were set up similarly to Section 3.4 with a couple of exceptions. First, the input $X$ includes the normalized spectrogam, log-mel deltas, and delta-deltas. Secondly our model was trained using Stochastic Gradient descent and warm restart similar to the setting from [24] for 126 epochs. Mix-up augmentation [27] was employed during our training. The baseline ResNet and compressed ResNet are independently trained and tested on the validation set for 10 times.

### 4.5. Performance

Table 6 presents the performance of different models on the DCASE 2021 Task 1A development dataset. The baseline CNN network is included as a reference. Note that the baseline CNN network and the compressed ResNet both required less than 60,000 parameters, while the baseline ResNet uses more than 360,000 parameters. The baseline CNN has very similar log loss measure to the ResNet models but its accuracy is significantly lower than the ResNet ones. The performance results showed that our compressed ResNet has a reduction of around 1.5% in accuracy as compared to the baseline ResNet; however, the compressed ResNet performs significantly better than the DCASE 2021 baseline model. In terms of log loss metric, the difference between the baseline ResNet and the compressed ResNet is very small. This suggests that our time-frequency separable convolution can

replace the 2D Convolution operations inside a complex model in order to significantly reduce the number of parameters both before and after training with only a small reduction in performance in a complex audio classification problem.

**Table 6.** Performance of the models on DCASE2021 Task 1A dataset.

| System | Accuracy (%) | Log Loss |
| --- | --- | --- |
| DCASE2021 Task 1A Baseline | $47.7 \pm 0.9$ | $1.473 \pm 0.05$ |
| Baseline ResNet | $65.99 \pm 0.12$ | $1.4700 \pm 0.0037$ |
| Compressed ResNet | $64.65 \pm 0.35$ | $1.4958 \pm 0.0033$ |

## 5. Discussions and Conclusions

All classes in the datasets in our experiments are balanced; therefore the classification accuracy is a reasonable performance metric. From the experimental results, we can conclude that in many acoustic scene classifications, utilizing the proposed time-frequency separable convolution structure can lead to a neural network model offering high performance while requiring many fewer parameters. In a simple audio classification task, the proposed time-frequency separable convolution structure actually led to improvement in performance in both log loss and accuracy. Given a well-performing model in a complex audio classification task, simply replacing convolutional layers by the time-frequency separable convolutions leads to at least 6-fold decrease in the number of parameters with only small differences in the classification performance. As a result, the time-frequency separable convolution structure is a promising configuration for learning audio features in audio classification applications. In addition, we also show that network architectures employing time-frequency separable convolution can combine with mix-up data augmentation for additional performance improvement.

The correlation between frequency components at a given time and the patterns of frequencies over a time window in spectrograms are essential features for developing audio classifiers. We think the proposed time-frequency separable convolution structure forces the networks to capture these features through the convolutions along frequency and time axes. Therefore, it significantly reduces the number of parameters while maintaining the high performance of the networks in acoustic scene classification. In addition, our proposed network also suggests that if 1D convolutions are configured properly, it can be very efficient to design low complexity solutions in audio applications.

We approached the low-complexity solution by designing a general model with fewer parameters; therefore, other techniques for compressing models such as pruning can still be applied on top of our proposed model for further reduction in size. An extension of our work could explore the combination of time-frequency separable convolution with pruning techniques to create a model which is smaller in size but still performs well. The time-frequency separable convolution may benefit other audio-oriented machine-learning applications by directly exploiting the time and frequency characteristic of the audio domain.

**Author Contributions:** Conceptualization, D.H.P. and D.L.J.; methodology, D.H.P.; software, D.H.P.; validation, D.H.P. and D.L.J.; formal analysis, D.H.P. and D.L.J.; investigation, D.H.P. and D.L.J.; resources, D.H.P. and D.L.J.; data curation, D.H.P. and D.L.J.; writing—original draft preparation, D.H.P.; writing—review and editing, D.L.J.; visualization, D.H.P.; supervision, D.L.J. All authors have read and agreed to the published version of the manuscript.

**Funding:** This work utilizes resources supported by the National Science Foundation's Major Research Instrumentation program, Grant #1725729, as well as the University of Illinois at Urbana-Champaign.

**Institutional Review Board Statement:** Not applicable.

**Informed Consent Statement:** Not applicable.

**Data Availability Statement:** Not applicable.

**Acknowledgments:** This research was supported by the ECE department at University of Illinois at Urbana-Champaign. The research is partially funded by the William L. Everitt Distinguished Professor endowment.

**Conflicts of Interest:** The authors declare no conflict of interest.

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
