# Peer review of "Low-Complexity Acoustic Scene Classification Using Time Frequency Separable Convolution"

_electronics, doi:10.3390/electronics11172734_

Round 1

Reviewer 1 Report

Deep neural network classifiers for acoustic scenes, which present reduced computational complexity through the use of separable time-frequency convolutions, are proposed. Classification results are similar to the state of the art, with a significant reduction in the number of network parameters.

For a better understanding of the development of the work, the methodology used to select the parameters of the proposed networks must be better described. For instance, it should be discussed how the filter sizes and the number of channels employed in the conv_tf blocks were obtained. 

Regarding the experimental results, more details should be included (such as the confusion matrix) to better understand the behavior of the proposed and baseline structures. 

Finally, more insights into the characteristics of the acoustic scenarios learned by the different classifiers would bring relevant contributions to the area.

Author Response

In our study, we tried to minimize the changes that we made to the original networks in order to present a fair and unbiased comparison of the performance of our proposed structure with full 2D convolution . We thus feared that an extensive search for the parameters might bias the results in favor of our proposed method or the particular application instead of highlighting the fact that our proposed structure could replace the many 2D convolutions in acoustic scene applications.

We provided extra details for the datasets and the set-up of the training set and test set for each experiment. In addition, we revised the conclusion to highlight the goals of the paper. Our goal is to show that the proposed structure is very promising for designing low complexity networks in acoustic scene applications, which is why we are using the accuracy metric as our performance measure.

Reviewer 2 Report

I do not have any suggestion

Author Response

Since the reviewer did not have any suggestions. We tried to revise our manuscript based on other feedbacks and suggestions. The changes are highlighted in the new version.

Reviewer 3 Report

This paper proposed a very interesting idea for audio signal analysis. The cascaded horizontal (time) and vertical (frequency) 1-d convolutions is quite helpful in reduction of model parameters and computation time when compared to 2-d convolutions, as the authors insisted. The idea is somewhat similar to 2-d Fourier transform on images, which, applying Fourier transforms along x-axis and then along y-axis to obtain equivalent 2-d Fourier transform of the raw images. One of the drawback is that the idea is too simple and no theoretical background is given why the performance is not degraded. However, if the main purpose of the authors is the reduction of the models, with the support of the experimental results with diverse types of CNN models, the reviewer agrees to it and it is convincing. 

In addition, the combination of 1-d convolutions is already taken in EEG applications, called EEGNet:

Q-EEGNet: an Energy-Efficient 8-bit Quantized Parallel EEGNet Implementation for Edge Motor-Imagery Brain--Machine Interfaces, Tibor Schneider, Xiaying Wang, Michael Hersche, Lukas Cavigelli, Luca Benini

https://www.catalyzex.com/paper/arxiv:2004.11690

The authors need to refer to this work and add the comparison to this model in the main text.

Author Response

We demonstrated through our experimental results that the proposed structure greatly reduces model complexity for audio application. We revised the conclusion to explain why our proposed method may work in audio contexts.

We have included some reviews and comparisons with EEGNet in the main text as suggested by the reviewer.

Reviewer 4 Report

The authors present a resnet architecture, where the 2D convolution is replaced by separable convolution. Since their inputs are spectrograms and their derivatives, the separable convolution can be explained as time-frequency separable convolution. Separable convolutions have been proposed before. However, the reviewer has not come across their application on spectrograms. In addition, the reviewer has not come across their application on this problem on audio scene classification, which attracts the interest of many scientists over the last years through the DCASE competitions.

The authors highlight that the use of separable convolution reduces the computational complexity of the resnet framework, reducing the performance only by a small percentage.

The paper is sound but of limited interest to the readers. One major correction that should be added to the paper is the following. Since the DCASE datasets are very unbalanced, the use of accuracy is not indicative of the overall performance. F1-score and mIoU are more representative scores that should be added to demonstrate that the performance of the network is not degraded too much by the introduction of the separable convolution.

After addressing this issue, the reviewer believes that the paper can be published.

Author Response

We address the comments of the reviewer by describing the experiments in more detail. We emphasize the fact the datasets that we are using for our experiments are balanced, and we keep the training set and test sets balanced. Therefore, it makes our selection of accuracy indicative of the overall performance. In addition, this is the reason the DCASE challenge on these datasets used accuracy to measure performance.

Round 2

Reviewer 4 Report

The authors have addressed my concerns, thus I recommend the acceptance of the paper.